# Efficacy and Biomarker Analysis of Adavosertib in Differentiated Thyroid Cancer

**DOI:** 10.3390/cancers13143487

**Published:** 2021-07-12

**Authors:** Yu-Ling Lu, Ming-Hsien Wu, Yi-Yin Lee, Ting-Chao Chou, Richard J. Wong, Shu-Fu Lin

**Affiliations:** 1Department of Internal Medicine, New Taipei Municipal TuCheng Hospital (Built and Operated by Chang Gung Medical Foundation), New Taipei City 23652, Taiwan; smartlynn18@cgmh.org.tw (Y.-L.L.); b9502013@cgmh.org.tw (M.-H.W.); winnielee@cgmh.org.tw (Y.-Y.L.); 2Department of Internal Medicine, Chang Gung Memorial Hospital, Taoyuan 33305, Taiwan; 3College of Medicine, Chang Gung University, Taoyuan 33302, Taiwan; 4Laboratory of Preclinical Pharmacology Core, Memorial Sloan-Kettering Cancer Center, New York, NY 10065, USA; dtchou99@gmail.com; 5Department of Surgery, Memorial Sloan-Kettering Cancer Center, New York, NY 10065, USA; wongr@mskcc.org

**Keywords:** adavosertib, Wee1, dabrafenib, trametinib, sorafenib, lenvatinib, combination therapy, differentiated thyroid cancer

## Abstract

**Simple Summary:**

Adavosertib is a first-in-class Wee1 inhibitor that has demonstrated activity against certain cancers. We evaluated the effects of adavosertib in treating differentiated thyroid cancer (DTC) using four DTC cell lines (BHP7-13, K1, FTC-133, FTC-238). Adavosertib accumulated cells in the G2/M phase and induced apoptosis in four DTC cell lines. Baseline Wee1 levels correlated with adavosertib sensitivity. Single-agent adavosertib therapy was sufficient to inhibit the growth of K1 and FTC-133 tumor models. Adavosertib potentiated the anti-tumor effect of dabrafenib and trametinib in K1 xenografts harboring the *BRAF^V^*^600E^ mutation, with promising safety profiles. Adavosertib also improved the anti-tumor efficacy of lenvatinib in FTC-133 xenografts. Our results suggest the clinical efficacy of adavosertib for DTC patient therapy.

**Abstract:**

Differentiated thyroid cancer (DTC) patients are usually known for their excellent prognoses. However, some patients with DTC develop refractory disease and require novel therapies with different therapeutic mechanisms. Targeting Wee1 with adavosertib has emerged as a novel strategy for cancer therapy. We determined the effects of adavosertib in four DTC cell lines. Adavosertib induces cell growth inhibition in a dose-dependent fashion. Cell cycle analyses revealed that cells were accumulated in the G2/M phase and apoptosis was induced by adavosertib in the four DTC tumor cell lines. The sensitivity of adavosertib correlated with baseline Wee1 expression. In vivo studies showed that adavosertib significantly inhibited the xenograft growth of papillary and follicular thyroid cancer tumor models. Adavosertib therapy, combined with dabrafenib and trametinib, had strong synergism in vitro, and revealed robust tumor growth suppression in vivo in a xenograft model of papillary thyroid cancer harboring mutant *BRAF*^V600E^, without appreciable toxicity. Furthermore, combination of adavosertib with lenvatinib was more effective than either agent alone in a xenograft model of follicular thyroid cancer. These results show that adavosertib has the potential in treating DTC.

## 1. Introduction

Differentiated thyroid carcinoma (DTC) comprises two major forms: papillary thyroid carcinoma (PTC) and follicular thyroid carcinoma (FTC) [1]. Its incidence has increased over the past four decades and accounts for approximately 90% of all cases of thyroid neoplasm [2]. Standard treatment for DTC includes surgery, radioactive iodine, and thyroid hormone therapy [3]. Most DTC patients exhibit excellent long-term survival following treatment. However, certain patients develop refractory disease and eventually die from it [4]. Sorafenib and lenvatinib, two multikinase inhibitors, were approved in 2013 and 2015, respectively, by the United States Food and Drug Administration (FDA) for progressive radioiodine-refractory DTC patients [3]. However, these drugs frequently induce some adverse side events that lead to dose reduction or treatment discontinuation [5]. In addition, disease progression developed 10.8 and 18.3 months after initiating sorafenib and lenvatinib therapy, respectively [6,7]. These observations mandate the search for novel therapies with different mechanisms against DTC.

Genomic instability is a feature of cancer resulting from a high rate of cell proliferation, exogenous and endogenous stressors, and compromised DNA damage response (DDR) [8,9], a complicated network comprising DNA damage repair and cell cycle checkpoint pathways [9]. Cancer cells typically lose one or more DDR pathways during oncogenesis and depend on the remaining DDR pathways to maintain cell viability in the face of genetic damage [10]. Targeted therapy based on DDR inhibition might be a promising therapeutic strategy [11].

Wee1, a protein kinase, regulates DNA damage repair and cell cycle [11]. Wee1 inhibition increases levels of p-H2AX (Ser139), a marker of DNA double-strand breaks [12]. The Wee1 mechanisms for maintaining genetic stability include inhibiting Mus81/Eme1 endonuclease activity, reducing replication initiation, preventing nucleotide shortage, and initiating homologous recombination repair [12,13,14]. Besides, Wee1 plays a critical role in the G2/M checkpoint that prevents entry into mitosis upon DNA damage. Entry into mitosis is regulated by the phosphorylation status of cyclin-dependent kinase 1 (CDK1) [15]. Inhibitory phosphorylation of CDK1 on tyrosine 15 is maintained by Wee1 before mitotic entry. Targeting Wee1 might be a therapeutic strategy for treating cancer [15].

Adavosertib (also known as AZD1775, MK-1775,) is a potent inhibitor of Wee1 with a median-effect dose (IC_50_) of 5.2 nmol/L [16]. Adavosertib has shown anti-tumor activity in monotherapy and combination with chemotherapy, ATR inhibitor, and poly (ADP-ribose) polymerase (PARP) inhibitor against malignancies [16,17,18,19,20]. The single-agent activity of adavosertib for fallopian tube carcinoma and squamous cell carcinoma with *BRCA* mutations has been demonstrated in a phase I clinical study [21]. A combinatorial approach with adavosertib plus carboplatin revealed therapeutic efficacy, including 5% complete remission and 43% partial response, in patients with *TP53*–mutated ovarian cancer resistant to prior carboplatin plus paclitaxel therapy, with manageable toxicity profiles [22]. In this study, we report the effects of adavosertib therapy for DTC.

## 2. Results

### 2.1. Adavosertib Induced Cytotoxicity in DTC Cell Lines

We evaluated the cytotoxic effects of adavosertib in four established DTC cell lines, including two PTC (BHP7-13 and K1) and two FTC (FTC-133 and FTC-238) cell lines. Adavosertib reduced cell survival across all cell lines in a dose-dependent fashion (Figure 1A) and, at 1000 nmol/L, inhibited cell growth by 91.6% (BHP7-13), 98.5% (K1), 98.0% (FTC-133), and 99.8% (FTC-238) on day 4. The adavosertib concentration leading to median-effect cytotoxicity (IC_50_) in the four DTC cell lines was determined on day 4 using CompuSyn software (Figure 1B) [23,24]. All DTC cell lines were sensitive to adavosertib, with IC_50_ values ranging from 71.8 to 175.6 nmol/L, which were below the achievable plasma concentration (1380–1650 nmol/L) in clinical trials [21,22].

### 2.2. Adavosertib Alters the Levels of p-CDK1, p-CHK1, and p-H2AX

Wee1 inhibition by adavosertib has been shown to reduce p-CDK1 (Tyr15) levels in pancreatic tumors [17]. We evaluated p-CDK1 (Tyr15) expression after 24 and 48 h of adavosertib (500 nmol/L) treatment by western blotting in four DTC cell lines. Adavosertib significantly reduced p-CDK1 (Tyr15) levels by 24 h, and the inhibitory effects persisted for 48 h in the BHP7-13, K1, FTC-133, and FTC-238 cells (Figure 1C). Adavosertib therapy has been shown to increase p-CHK1 (Ser345) and p-H2AX (Ser139) levels, two markers of DNA damage in cancer cells [16]. We found that adavosertib (500 nmol/L) increased the p-CHK1 (Ser345) and p-H2AX (Ser139) levels by 24 h in the four DTC cell lines.

### 2.3. Adavosertib Induced Cell Cycle G2/M Phase Arrest

We examined the effect of adavosertib (500 nmol/L) on cell cycle progression in the four DTC cell lines. BHP7-13 cell line revealed the effect of adavosertib on cell cycle distribution (Figure 2A). Besides, we analyzed the cell cycle data following adavosertib therapy, which showed an increase in the G2/M phase in the four DTC cell lines (Figure 2B). Adavosertib significantly induced cells accumulation in the G2/M phase when compared with the control treated BHP7-13 (19.5% ± 0.3% and 11.2% ± 0.2%, *p* < 0.001), K1 (39.6% ± 0.2% and 7.6% ± 0.2%, *p* < 0.001), FTC-133 (34.3% ± 0.1% and 16.7% ± 0.3%, *p* < 0.001), and FTC-238 (74.4% ± 0.1% and 21.1% ± 0.2%, *p* < 0.001).

Prior studies have shown that adavosertib treatment led to mitotic arrest in pancreatic cancer and lung cancer cells [17,25], although this effect was not observed in most (80%) of the human tumor specimens treated with adavosertib [21]. We examined adavosertib’s ability to accumulate DTC cells in the mitotic phase. Figure 2C shows a representative BHP7-13 cell line. Mitotic cells were identified, and the mitotic index was calculated for the four DTC cell lines (Figure 2D). Adavosertib treatment did not significantly change the percentage of mitotic cells in BHP7-13 when compared with the placebo treated cells (1.5% ± 0.2% and 1.4% ± 0.1%, *p* = 0.625). However, adavosertib (500 nmol/L) treatment significantly reduced the accumulation of mitotic cells in K1 (0.8% ± 0.1% and 1.1% ± 0.1%, *p* = 0.005), FTC-133 (0.0% ± 0.0% and 1.2% ± 0.1%, *p* < 0.001), and FTC-238 (1.9% ± 0.2% and 3.0% ± 0.3%, *p* = 0.008), indicating that adavosertib therapy inhibited mitotic entry in the K1, FTC-133, and FTC-238 cell lines. These data suggest that adavosertib treatment did not induce mitotic arrest in the DTC cell lines.

We also calculated the proportions of cells with p-Histone H3 staining in the BHP7-13, K1, FTC-133, and FTC-238 cells lines (Figure 2E). Adavosertib significantly increased the fraction of cells with p-Histone H3 staining in the BHP7-13 (2.5% ± 0.2% and 2.1% ± 0.1%, *p* = 0.046) and FTC-133 (5.3% ± 0.8% and 3.3% ± 0.5%, *p* = 0.042) cell lines but not in the K1 (6.0% ± 0.5% and 4.2% ± 0.5%, *p* = 0.206) and FTC-238 (6.1% ± 1.3% and 3.8% ± 0.8%, *p* = 0.155) cell lines. A prior report demonstrated that phosphorylation of histone H3 appears in the late G2 and mitosis phases [26].

### 2.4. Effects of Adavosertib on Apoptosis

Adavosertib treatment induces apoptotic cell death in leukemia cells [27]. Caspase-3 executes apoptosis following the activation of extrinsic and intrinsic pathways [28]. We evaluated the effect of adavosertib therapy on caspase-3 activity in all four DTC cell lines using a fluorometric assay, and the data was presented as optical densities (OD) (Figure 3A). When compared with placebo, adavosertib increased caspase-3 activity in BHP7-13 (0.022 ± 0.000 OD and 0.018 ± 0.000 OD, *p* = 0.008), K1 (0.097 ± 0.000 OD and 0.077 ± 0.001 OD, *p* = 0.001), FTC-133 (0.042 ± 0.000 OD and 0.020 ± 0.001 OD, *p* = 0.001), and FTC-238 (0.031 ± 0.000 OD and 0.017 ± 0.001 OD, *p* = 0.003) DTC cell lines.

Caspase-3 activation can contribute to apoptotic cell death. We assessed whether adavosertib treatment could induce early apoptosis in the four DTC cell lines using Annexin V-Alexa Fluor 488 and propidium iodide (PI) staining (Figure 3B). The statistical analysis demonstrated that adavosertib (500 nmol/L) significantly increased the pro-portion of early apoptotic cells in BHP7-13 (4.1% ± 0.1% and 1.8% ± 0.1%, *p* < 0.001), K1 (8.4% ± 0.1% and 0.6% ± 0.1%, *p* < 0.001), FTC-133 (5.0% ± 0.5% and 1.1% ± 0.1%, *p* = 0.002), and FTC-238 (3.4% ± 0.0% and 0.9% ± 0.1%, *p* < 0.001) when compared with the placebo (Figure 3C).

Using flow cytometry, we measured the efficacy of adavosertib (500 nmol/L) to induce sub-G1 apoptosis in the four DTC cell lines (Figure 3D). We exposed DTC cells to adavosertib and calculated the proportion of sub-G1 cells (Figure 3E). Adavosertib treatment significantly increased the proportion of sub-G1 cells in BHP7-13 (1.6% ± 0.0% and 0.8% ± 0.0%, *p* < 0.001), K1 (3.0% ± 0.0% and 1.9% ± 0.1%, *p* < 0.001), FTC-133 (33.9% ± 0.3% and 5.0% ± 0.0%, *p* < 0.001), and FTC-238 (16.3% ± 0.1% and 2.6% ± 0.2%, *p* < 0.001), indicating that these DTC cells were undergoing apoptosis due to the adavosertib treatment.

To validate the induction of apoptosis in the DTC cells by adavosertib, we assessed cleaved PARP, a marker of apoptosis, by western blot 24 and 48 h after adavosertib treatment (500 nmol/L) (Figure 3F). Adavosertib increased the expression of cleaved PARP in BHP7-13 (48 h), K1 (24 h), FTC-133 (48 h), and FTC-238 (48 h). These findings are in line with prior published studies and suggest that apoptotic mechanisms account for the cytotoxicity of adavosertib in four DTC cell lines [18,29].

### 2.5. Adavosertib Sensitivity Correlates with Baseline Wee1 Expression

We recently determined the IC_50_ of adavosertib in three anaplastic thyroid cancer cell lines as follows: 180.1 nmol/L (KAT18), 303.4 nmol/L (8505C), and 373.0 nmol/L (8305C) [30]. The IC_50_ of adavosertib presents a wide range across seven cell lines of thyroid follicular cell origin, with a 5.2-fold difference between FTC-133 and 8305C. We assessed the potential biomarkers correlating with sensitivity to adavosertib in seven thyroid cancer cell lines. The expression of Wee1, PLK1, p-CDK1 (Tyr15), p-CHK1 (Ser345), AXL, cyclin E1, and Myt1 were evaluated in these cell lines (Figure 4A). Baseline Wee1, PLK1, p-CDK1 (Tyr15), and p-CHK1 (Ser345) levels were assessed, given that adavosertib, directly and indirectly, targets these proteins [29]. We selected AXL, cyclin E1, and Myt1 because these proteins have been implicated in the sensitivity of adavosertib [25,31,32]. We ordered the sensitivity of the seven cell lines with adavosertib according to the IC_50_ and observed that Wee1 expression gradually increased with the IC_50_. The statistical relationships analyzed with Pearson’s correlation coefficient (Figure 4B) showed that Wee1 expression had a significant correlation (*R* = 0.871, *p* = 0.011) with adavosertib sensitivity. The expression of baseline PLK1, p-CDK1 (Tyr15), p-CHK1 (Ser345), AXL, cyclin E1, and Myt1 revealed no correlation with adavosertib sensitivity.

### 2.6. Interaction between Adavosertib and Targeted Therapies in DTC Cells

We analyzed the effects of adavosertib combined with sorafenib against DTC cells. The cytotoxic effects and IC_50_ of sorafenib in four DTC cell lines were evaluated in this (Appendix A) and prior studies [33], and the resulting data were employed for the combination therapy study of adavosertib and sorafenib. The combination therapy of adavosertib and sorafenib significantly improved cytotoxicity in the four DTC cell lines over single-agent therapy (Figure 5A). Interactions between adavosertib and sorafenib were determined by calculating the combination index (CI) using the Chou–Talalay equation (Figure 5B) [23,24]. The combination of adavosertib and sorafenib was synergistic in K1 (CI, 0.72–0.73), and ranged from synergistic to antagonist in BHP7-13, FTC-133, and FTC-238 (CI of 0.66–1.06, 0.97–1.02, and 0.53–1.07, respectively). The results demonstrate that the adavosertib and sorafenib combination was mostly synergistic in treating DTC cells.

We also assessed the therapeutic effects of the combination of adavosertib and lenvatinib in the four DTC cell lines by determining the dose-response curves and IC_50_ of lenvatinib in the four DTC cell lines (Appendix AA,B). The results were used to study the combination of adavosertib and lenvatinib, which had increased cytotoxicity over therapy with either agent alone (Figure 5C). The Chou–Talalay Combination index was employed to evaluate the drug-drug interaction between adavosertib and lenvatinib and showed that the combination was synergistic in FTC-133 (CI, 0.43–0.67) and FTC-238 (CI, 0.53–0.96) and mostly synergistic in BHP7-13 and K1 (fraction affected ≥ 0.4) (Figure 5D). The results indicate that the adavosertib and lenvatinib combination was mainly synergistic in DTC cells.

*BRAF*^V600E^ mutation is a common genetic alteration (45.7%) in PTC and appears less frequently (1.4%) in FTC [34]. It is associated with increased cancer-related mortality in PTC patients [35]. Dual BRAF and MEK inhibition therapy has emerged as a novel treatment alternative for patients with *BRAF* mutations; however, acquired resistance remains a concern [36]. The US FDA recently approved the combination of dabrafenib (a BRAF inhibitor) and trametinib (a MEK inhibitor) for treating *BRAF*^V600E^-mutated anaplastic thyroid cancer [37]. We assessed the therapeutic effects of adavosertib combined with dabrafenib and trametinib in K1 cells harboring the *BRAF*^V600E^ mutation [38] by determining the dose-response curves and the IC_50_ of dabrafenib and trametinib in K1 cells (Appendix AA,B). The results were employed to study the triple combination of adavosertib and dabrafenib plus trametinib, which increased cytotoxicity over the single modality treatment (Figure 5E). The triple combination treatment showed synergism in K1 cells (CI, 0.50–0.82) (Figure 5F).

### 2.7. Adavosertib Therapy Shows Anti-Tumor Efficacy in DTC Xenograft Models

With the in vitro promising results, we examined the in vivo effect of adavosertib monotherapy and combination with targeted therapies. Nude mice were inoculated with K1 cells in the right flank. Once the tumors reached a mean diameter of 7.0 mm, the mice were treated with vehicle, adavosertib (50 mg/kg), sorafenib (40 mg/kg), adavosertib (50 mg/kg) and sorafenib (40 mg/kg), dabrafenib (30 mg/kg) plus trametinib (0.6 mg/kg), and the triple combination of adavosertib (50 mg/kg), dabrafenib (30 mg/kg), and trametinib (0.6 mg/kg) (*n* = 4 per group) (Figure 6A). Adavosertib, sorafenib, and dabrafenib plus trametinib significantly reduced tumor growth versus vehicle treatment (*p* < 0.001 for both comparisons). The combination of adavosertib and sorafenib and the triple combination of adavosertib, dabrafenib, and trametinib did not significantly improve the therapeutic efficacy over sorafenib and dabrafenib plus trametinib (*p* = 0.998 and *p* = 0.997, respectively). However, the triple combination of adavosertib, dabrafenib, and trametinib resulted in complete remission in 25% (1 of 4) of the K1 tumors between days 18 and 21, at which point the study was closed due to the control animals being close to the humane endpoints. This 25% complete response rate demonstrates significant promise, and the effect was not observed in the other five treatment groups. There was no obvious weight loss following multiple treatments in the treatment groups during the entire study (Figure 6B). A group of mice photographed on day 15 revealed tumor dimension reduction following the drug treatment (Figure 6C). We performed additional experiments to confirm the therapeutic efficacy of triple combination of adavosertib, dabrafenib, and trametinib in K1 xenografts (Appendix AA). Triple combination therapy led to complete tumor regression in 60% (3/5) of the K1 tumors between days 14 and 21 after two cycles of treatment. There was no obvious weight change between the dual and triple combination therapy (Appendix AB). These data strengthen our primary observation that the triple combination of adavosertib, dabrafenib, and trametinib had robust therapeutic effects against K1 tumors.

Nude mice bearing FTC-133 tumors with a mean diameter of 5.7 mm were treated with vehicle (*n* = 6), adavosertib (50 mg/kg, *n* = 6), sorafenib (40 mg/kg, *n* = 5), and combination therapy (*n* = 6) daily for three cycles of 5-days-on and 2-days-off treatment (Figure 7A). Adavosertib, sorafenib, and adavosertib plus sorafenib significantly retarded FTC-133 tumor growth as compared with the vehicle treatment (*p* < 0.001 for both comparisons). The combination therapy of adavosertib and sorafenib did not significantly repress FTC-133 tumor growth compared with either single-agent treatment. When the study was closed, there was no significant reduction in body weight with the adavosertib, sorafenib, or combination therapy compared with placebo treatment (Figure 7B). Representative nude mice with FTC-133 tumors were photographed on day 21 (Figure 7C).

We also investigated the in vivo effect of adavosertib and lenvatinib in the FTC-133 xenograft model. Nude mice, inoculated with FTC-133 cells in the right flank, were treated with vehicle, adavosertib (50 mg/kg), lenvatinib (30 mg/kg), and the combination of adavosertib (50 mg/kg) and lenvatinib (30 mg/kg) (*n* = 5 per group), daily, for three cycles of 5-days-on and 2-days-off treatments (Figure 7D) once the tumors reached a mean diameter of 5.6 mm. Adavosertib, lenvatinib, and adavosertib plus lenvatinib significantly reduced tumor growth versus the control treatment (*p* = 0.002, *p* = 0.003, and *p* < 0.001, respectively). The combination of adavosertib and lenvatinib significantly improved the therapeutic efficacy versus adavosertib and lenvatinib alone (*p* < 0.001 and *p* = 0.011, respectively). There was no obvious weight loss in the treatment groups compared with the control group during the study period (Figure 7E). A group of mice photographed on day 21 demonstrated tumor size reduction after the drug treatment (Figure 7F).

To evaluate the molecular effects of a single adavosertib (50 mg/kg) treatment in K1 and FTC-133 xenografts (Appendix A), we performed immunoblotting, which showed that p-CDK1 (Tyr15) was reduced by 4 h in the K1 and FTC-133 tumors. The p-CHK1 (Ser345) expression was reduced in the K1 (24 h), but increased in the FTC-133 tumor (4 h). p-H2AX (Ser139) levels were increased in the K1 (4 and 8 h) and FTC-133 (8 h) tumors. Proliferating cell nuclear antigen (PCNA) levels, a marker of cell proliferation, were not significantly changed in the K1 and FTC-133 tumors during the study period. Cleaved caspase-3 levels were enhanced in the K1 and FTC-133 (by 8 h) tumors.

## 3. Discussion

In this study, we aimed at evaluating the therapeutic efficacy of adavosertib monotherapy and the combination of adavosertib and kinase inhibitors against DTC. Adavosertib efficiently reduced cell viability in four DTC cell lines, and adavosertib monotherapy significantly inhibited xenograft growth in PTC and FTC tumor models. Adavosertib potentiated the therapeutic effects of dabrafenib and trametinib in the K1 tumor model, without appreciable toxicity. The complete response rate of 25–60% in K1 xenografts in the triple combination therapy group compared with 0% in the other treatment groups is also notable. Adavosertib also improved the antitumor effect of lenvatinib in the FTC-133 model. These study findings suggest that adavosertib has the potential for treating patients with DTC.

Adavosertib accumulated the DTC cells in the G2/M phase, which might be one of the drug’s treatment mechanisms for DTC. Adavosertib inhibited DTC cells in the G2 phase (instead of the M phase) in four DTC cell lines. Adavosertib induced a higher proportion of cells in the G2/M phase but did not increase the proportion of mitotic cells, demonstrating that adavosertib accumulates DTC cells in the G2 phase. CDK1 activation is essential for commitment to mitosis, and CDK1 activity is regulated by Wee1, Myt1, and CDC25C [39]. Wee1 inhibition by adavosertib might be insufficient to fully activate CDK1 for mitosis in DTC cells.

Adavosertib is a potent PLK1 inhibitor, with IC_50_ at nanomolar concentrations (22–34 nmol/L) [40]. PLK1 is involved in numerous biological functions, including mitotic entry and cell survival [41,42]. PLK1 inhibition by adavosertib treatment might delay/prevent mitotic entry and induce apoptosis in DTC cells.

We employed two DTC xenograft models in this study, and adavosertib consistently inhibited tumor growth in both tumor models. These results demonstrate that single-agent adavosertib is effective for DTC therapy. The addition of adavosertib significantly improved the therapeutic efficacy of lenvatinib in a FTC tumor model. However, the therapeutic effect of adavosertib and sorafenib combination was not greater than that of single sorafenib therapy. An optimization of treatment regimen might improve the curative effects of the adavosertib and sorafenib combination therapy in DTC xenografts.

Adavosertib sensitivity was associated with lower Wee1 protein levels in thyroid cancer cell lines. Our data suggest that thyroid cancer cell lines with lower Wee1 expression rely on Wee1 activity, and interrupting Wee1 with adavosertib impairs cell growth more significantly than thyroid cancer cells with higher Wee1 expression. This potential predictive biomarker might help in designing the patient selection process for clinical trials.

Adavosertib treatment increased p-CHK1 (Ser345) expression in FTC-133 tumors, but decreased p-CHK1 (Ser345) levels in K1 tumors. A recent report demonstrated that adavosertib treatment initially increased p-CHK1 (Ser345) expression (by 2–8 h), followed by gradually decreasing expression (by 16–72 h) in four lymphoma cell lines [43]. The expression of p-CHK1 indicates ATR-CHK1 pathway activation, while earlier p-CHK1 reduction is correlated with drug sensitivity.

There are numerous clinical studies using adavosertib as monotherapy and in combination with chemotherapy, with encouraging results revealing that these treatment regimens are generally tolerated [21,22,44,45,46]. Our study provides preclinical data that warrant clinical trials to examine adavosertib for DTC therapy.

## 4. Materials and Methods

### 4.1. Cell Lines and Cell Culture

We studied two PTC cell lines (BHP7-13 and K1) and two FTC cell lines (FTC-133 and FTC-238). The BHP7-13 cells have been previously described [33]. The K1, FTC-133, and FTC-238 cells were purchased from Sigma (now Merck, Darmstadt, Germany). All four cell lines were authenticated using short tandem repeat DNA profiling [47,48]. The BHP7-13 cells were maintained in Roswell Park Memorial Institute 1640 media supplemented with sodium bicarbonate (2.0 g/L). The K1 cells were maintained in Dulbecco’s Modified Eagle Medium, Ham’s F12, and MCDB 105 (2:1:1) with glutamine (2.0 mmol/L). The FTC-133 and FTC-238 cells were maintained in Dulbecco’s Modified Eagle Medium and Ham’s F12 (1:1) with glutamine (2.0 mmol/L). All media were supplemented with 10% fetal bovine serum, 100,000 units/L of penicillin, and 100 mg/L of streptomycin. All cells were maintained in standard tissue culture conditions (5% carbon dioxide humidified incubator at 37 °C).

### 4.2. Pharmacologic Agents

Adavosertib, sorafenib, lenvatinib, dabrafenib, and trametinib were purchased from Selleck Chemicals, and diluted in dimethyl sulfoxide (DMSO; Merck) to a concentration of 10 mmol/L and stored at −80 °C until in vitro use. For the in vivo experiments, adavosertib and lenvatinib were diluted in methyl cellulose (Merck) and distilled water (1:200 *w*/*v*) to a final concentration of 12 mg/mL. Sorafenib was dissolved in 50/50% Kolliphor EL (Merck) and ethanol (Merck) and further diluted with water to a final concentration of 14.4 mg/mL before use. Dabrafenib and trametinib were dissolved in 0.5% (hydroxypropyl) methylcellulose (Merck), 0.2% Tween 80 (Merck), and distilled water to concentrations of 8 mg/mL and 0.16 mg/mL, respectively. All drugs were stored at −80 °C until their in vivo use.

### 4.3. Antibodies

Antibodies targeting p-CDK1 (Tyr15), p-CHK1 (Ser345), p-H2AX (Ser139), p-Histone H3 (Ser10), PARP, Wee1, PLK1, AXL, cyclin E1, Myt1, PCNA, cleaved caspase-3, and β-actin were purchased from Cell Signaling Technology. Antibody to α-tubulin was purchased from Merck.

### 4.4. Cell Viability Assays and Drug Synergy Studies

DTC cells were cultured at a density of 2 × 10^3^ cells per well in 24-well plates in 1 mL of media and incubated overnight. We added six serial two-fold dilutions of adavosertib, sorafenib, lenvatinib, dabrafenib, trametinib, or vehicle for 4-day treatments before cell viability was determined. After removing the culture media, the cells were washed with PBS and lysed with Triton X-100 (1.35%, Merck) to release intracellular lactate dehydrogenase (LDH). Cell viability was assessed using an LDH assay kit (Promega, Madison, WI, USA) and quantified LDH levels using spectrophotometry (Infinite M200 PRO, Tecan, Männedorf, Switzerland) according to the manufacturer’s protocol. All experiments were performed in triplicate, and the results are shown as the percentage of cells normalized to the placebo samples, which were considered 100% viable. We determined IC_50_ using CompuSyn software for each cell line on day 4 [23,24].

To study the drug combinations of adavosertib and targeted therapy, cells were cultured in 1 mL media for overnight at 2 × 10^3^ cells per well in 24-well plates and cells were treated with vehicle, adavosertib and targeted therapy at a fixed-dose ratio, or combination therapy simultaneously for a 4-day course before evaluating cell viability. Six serial of two-fold dilutions were investigated at the following starting doses for BHP7-13, K1, FTC-133, and FTC-238, respectively: adavosertib at 702.4, 356.4, 287.2, and 383.2 nmol/L, sorafenib at 2.8, 18.0, 23.6, and 29.6 μmol/L, and lenvatinib at 0.2, 26.32, 11.84, and 29.96 μmol/L. To study the combination therapy of adavosertib and dabrafenib plus trametinib, K1 cells were incubated with vehicle, adavosertib, dabrafenib plus trametinib, or triple combination of adavosertib, dabrafenib, and trametinib simultaneously for 4-days. Six serial of two-fold dilutions were analyzed at the following starting doses for K1 cells: adavosertib at 356.4 nmol/L, dabrafenib at 1.0 nmol/L, and trametinib at 0.1 nmol/L for a 4-days course. These starting doses were determined using the IC_50_ obtained in this and prior studies [33]. For the drug combinations, the Chou–Talalay method and CompuSyn software (Paramus, NJ, USA) were used to calculate the quantitative CI that determined an additive effect (CI = 1), synergy (CI < 1), and antagonism (CI > 1).

### 4.5. Western Blot Analysis

DTC cells were treated with adavosertib (500 nmol/L) or vehicle for 24 and 48 h after plating overnight the cells at 1 × 10^6^ cells in 100-mm petri dishes in 10 mL of media. Cell pellets were dissolved using immunoprecipitation lysis buffer containing protein phosphatase inhibitor mixture (Bionovas, Toronto, ON, Canada), sonicated, and clarified by centrifugation. Equal amounts of protein lysate were separated by 12% Tris-HCl gels, transferred to polyvinylidene difluoride membranes, blocked with 5% fat-free milk, and exposed to the primary antibody followed by a secondary antibody conjugated to horseradish peroxidase. Proteins were detected by an enhanced chemiluminescence kit (PerkinElmer, Waltham, MA, USA) using UVP ChemStudio PLUS touch (Analytik Jena, Jena, Germany).

Band densitometry was performed using Molecular Imager VersaDoc MP 4000 system software (Bio-Rad, Hercules, CA, USA). The ratios of Wee1, PLK1, p-CDK1 (Tyr15), p-CHK1 (Ser345), AXL, cyclin E1, and Myt1 to β-actin were calculated in each cell line to determine the relative expression using untreated FTC-133 cell values as reference. Original western blot images could be viewed in File S1.

### 4.6. Flow Cytometry Analysis for Cell Cycle

To elucidate the effect of adavosertib on cell cycle distribution, we seeded cells overnight at 4 × 10^5^ cells (BHP7-13, K1, FTC-133) or 1 × 10^5^ cells (FTC-238) per well in 6-well plates in 2 mL of media. Adavosertib (500 nmol/L) or vehicle was added, and the cells were incubated for 48 h (BHP7-13, K1, FTC-133) or 24 h (FTC-238). Adherent cells were then trypsinized and collected, washed with PBS, fixed with ice-cold 70% ethanol, washed again with PBS, incubated with RNase A (100 μg/mL) (Merck), and stained with PI (5 μg/mL) (Merck) at 37 °C for 15 min in the dark. Cell cycle distribution was acquired by flow cytometry (BD FACSCalibur Flow Cytometer, BD Biosciences, Franklin Lakes, United States). Each condition was performed in triplicate.

### 4.7. Apoptotic Cell Death Assessment

Caspase-3 activity induced by adavosertib treatment on the four DTC cell lines was determined using a fluorometric assay kit (Abcam, Cambridge, United Kingdom). Cells were seeded at 1.5 × 10^5^ cells (BHP7-13), 1.0 × 10^6^ cells (K1), or 2 × 10^5^ cells (FTC-133, FTC-238) in 100-mm petri dishes in 10 mL of media overnight. Adavosertib (500 nmol/L) or placebo was added and incubated for 48 h (BHP7-13, K1) or 24 h (FTC-133, FTC-238). Adherent cells (5 × 10^5^) were collected, centrifuged, lysed on ice for 10 min using 50 µL of lysis buffer, and incubated at 37 °C for 1.5 h with caspase-3 substrate (DEVD-AFC) and reaction buffer. Spectro-photometry (Infinite M200 PRO, Tecan) was employed to assess caspase-3 activity, and the results are presented as OD. Each condition was performed in duplicate.

Annexin V-Alexa Fluor 488 and PI kit (Invitrogen, Waltham, MA, United States) was applied to determine early apoptosis. DTC cells were plated at a density of 2 × 10^5^ cells (BHP7-13, K1, FTC-133) or 4 × 10^5^ cells (FTC-238) in 6-well plates in 2 mL of media overnight, and the cells were treated with adavosertib (500 nmol/L) or placebo for 48 h (BHP7-13, K1, FTC-133) or 24 h (FTC-238). Adherent cells were collected, washed with PBS, and incubated with Annexin V-Alexa Fluor 488 and PI at room temperature for 15 min in the dark. Flow cytometry (BD FACSCalibur Flow Cytometer, BD Biosciences) was employed to detect early apoptotic cells (Annexin V-positive, PI-negative). Each condition was performed in triplicate.

Using flow cytometry, we studied the ability of adavosertib to induce sub-G1 apoptotic cells. DTC cells were plated at 4 × 10^5^ cells (BHP7-13, FTC-133), 2 × 10^5^ cells (K1), and 1 × 10^5^ cells (FTC-238) per well in 6-well plates in 2 mL of media overnight. Adavosertib (500 nmol/L) or vehicle was added, and the cells were incubated for 48 h (BHP7-13, K1, FTC-133) or 24 h (FTC-238). Floating and trypsinized adherent cells were collected and prepared as described for cell cycle evaluation. Flow cytometry detection of the apoptotic sub-G1 (BD FACSCalibur Flow Cytometer, BD Biosciences) was performed in triplicate, as measured by DNA content.

### 4.8. Immunofluorescence Microscopy

Confocal microscopy was performed to evaluate the effect of adavosertib on mitosis. DTC cells were plated for overnight at 5 × 10^4^ cells (BHP7-13, FTC-238) and 1 × 10^5^ cells (K1, FTC-133) in 4-well culture slides in 1 mL of media, and were incubated for 48 h (BHP7-13, K1, FTC-133) and 24 h (FTC-238) in the presence of adavosertib (500 nmol/L) or placebo. The cells were then washed with PBS, fixed with 4% paraformaldehyde (Merck) for 15 min at room temperature, washed with PBS, permeabilized with 0.1% Triton X-100 for 10 min, and washed with PBS. The samples were then probed with primary rabbit p-Histone H3 (Ser10) antibody (1:200) and mouse α-tubulin antibody (1:1000) at 4 °C overnight in a humidified chamber. The slides, stained in the dark with secondary Alexa Fluor 633-conjugated goat anti-rabbit antibody (1:1000; Invitrogen) and Alexa Fluor 488-conjugated goat anti-mouse antibody (1:1000; Life Technologies, Carlsbad, CA, United States) for 25 min at 37 °C, were washed with PBS and stained with 4′,6-diamidino-2-phenylindole (DAPI; 0.2 μg/mL) (Invitrogen) for 10 min at room temperature, covered with mounting medium. Chromosomes were examined to identify mitotic cells. We also identified thyroid cancer cells with p-Histone H3 staining. Samples were imaged with a Leica TCS SP8 X confocal microscope (Leica Microsystems, Wetzlar, Germany).

### 4.9. In Vivo Flank Xenograft Tumor Therapy

The animals were handled following the protocol approved by the Animal Care and Use Committee at Chang Gung Memorial Hospital, Linkou, Taiwan (approved on 22 March 2019, permission no. 2019010202). K1 and FTC-133 flank tumors were established by injecting 1 million cells in 100 μL of extracellular matrix gel (Merck): culture medium (1:1) into the subcutaneous flanks of 8–9-week-old female athymic nude mice (National Laboratory Animal Center, Taiwan). These DTC cell lines were chosen because they are more tumorigenic in vivo.

Mice bearing K1 tumors underwent oral gavage treatment of placebo, adavosertib (50 mg/kg), sorafenib (40 mg/kg), adavosertib (50 mg/kg) plus sorafenib (40 mg/kg), dabrafenib (30 mg/kg) plus trametinib (0.6 mg/kg), and triple combination of adavosertib (50 mg/kg), dabrafenib (30 mg/kg), and trametinib (0.6 mg/kg) once a day for 5-days-on and 2-days-off for three cycles. Mice bearing FTC-133 tumors underwent oral gavage treatment of placebo, adavosertib (50 mg/kg), sorafenib (40 mg/kg), lenvatinib (30 mg/kg), adavosertib (50 mg/kg) plus sorafenib (40 mg/kg), and adavosertib (50 mg/kg) plus lenvatinib (30 mg/kg) once a day for 5-days-on and 2-days-off for three cycles. The doses for adavosertib, sorafenib, lenvatinib, and dabrafenib plus trametinib were chosen based on previous reports [16,49,50,51]. We measured the tumor diameters twice a week using an electronic caliper and calculated the tumor volumes using the following formula: a × b^2^ × 0.4, where a and b represent the tumor’s long and short diameters, respectively. We measured each mouse’s body weight twice a week as a marker of toxicity.

### 4.10. Statistical Analysis

The statistical analysis was conducted using SPSS statistical software (version 22.0, SPSS Inc., Chicago, IL, USA), employing Student’s t-test to compare two groups of data. Differences between more than two treatment groups were determined by two-way ANOVA followed by a post hoc Scheffe test. Results are expressed as mean ± standard error and all *p*-value of < 0.05 were considered statistically significant.

## 5. Conclusions

Our results reveal that adavosertib treatment induces cytotoxicity in DTC cells. Two DTC xenograft models showed the therapeutic efficacy and safety of adavosertib. Adavosertib potentiates the therapeutic efficacy of dabrafenib and trametinib in the K1 xenograft model and the therapeutic efficacy of lenvatinib in FTC-133 tumors. Our findings have important clinical implications for using adavosertib in treating DTC, as a single drug and in combination therapies. Further clinical investigations of adavosertib in treating this disease are indicated.

## Figures and Tables

**Figure 1 cancers-13-03487-f001:**
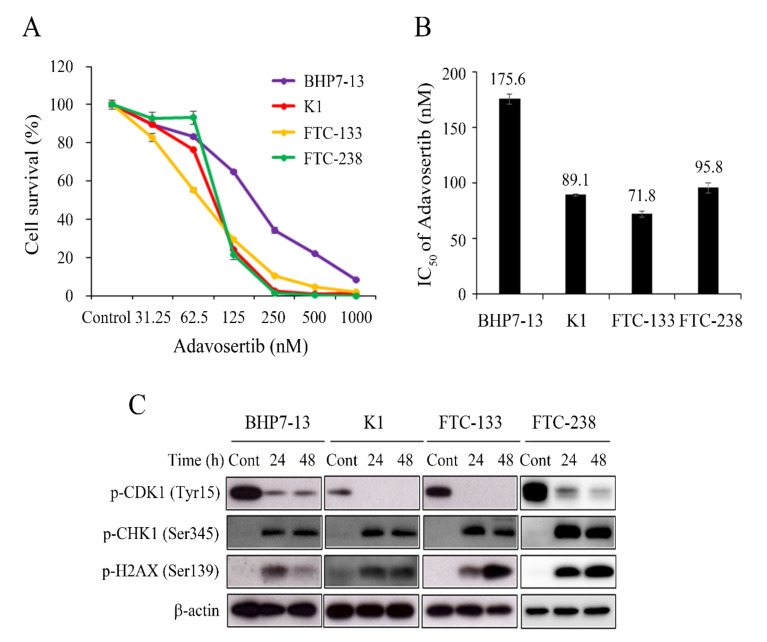
Adavosertib treatment induces dose-dependent cytotoxicity in four DTC cell lines. (**A**) Dose-response curves were obtained from cells treated with adavosertib for a 4-day treatment course in four DTC cell lines. (**B**) The median-effect dose (IC_50_) of adavosertib on day 4 was determined by CompuSyn software for each cell line. (**C**) DTC cells were treated with vehicle or adavosertib (500 nmol/L) for 24 and 48 h, after which immunoblotting was performed to determine the expression of p-CDK1 (Tyr15), p-CHK1 (Ser345), and p-H2AX (Ser139). p-CDK1 (Tyr15) expression was decreased, and p-CHK1 (Ser345) and p-H2AX (Ser139) expression was increased by adavosertib in the four cell lines.

**Figure 2 cancers-13-03487-f002:**
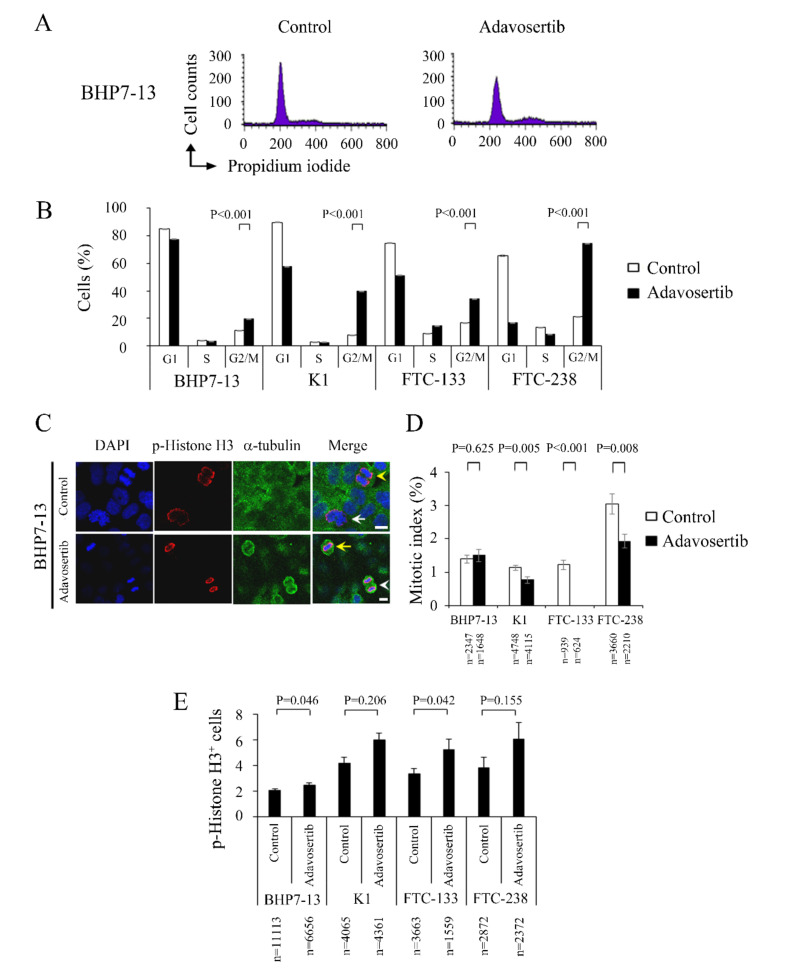
Adavosertib inhibits cell cycle progression in the G2/M phase in four DTC cell lines. (**A**) Cell cycle analysis measuring the DNA content of 1 × 10^4^ events using flow cytometry was performed in BHP7-13 cells treated with placebo or adavosertib (500 nmol/L) for 48 h. (**B**) Cell cycle analysis by measuring propidium iodide staining in DTC cells treated with vehicle or adavosertib (500 nmol/L) revealed that adavosertib accumulated cells in the G2/M phase by 48 h (BHP7-13, K1, FTC-133) and 24 h (FTC-238). (**C**) BHP7-13 cells were treated with adavosertib (500 nmol/L) or vehicle for 48 h and stained with fluorescent antibodies targeting DAPI (blue), p-Histone H3 (Ser10) (red) and α-tubulin (green). We examined the chromosome characteristics of the BHP7-13 cells using confocal microscopy. Cells in prophase (white arrow), metaphase (yellow arrow), anaphase (yellow arrowhead), and telophase (white arrowhead) are indicated. (**D**) The percentage of cells in mitosis was evaluated after treatment with vehicle or adavosertib (500 nmol/L) for 48 h (BHP7-13, K1, FTC-133) and 24 h (FTC-238). Cells were stained with DAPI, and chromosome features were evaluated using immunofluorescence confocal microscopy. The mitotic index was assessed with a minimum of 624 cells counted from at least 10 different fields for each condition. Adavosertib significantly reduced the proportion of cells in mitosis in K1, FTC-133, and FTC-238 but did not appreciably alter the proportion of mitosis in the BHP7-13 cells. (**E**) The percentage of DTC cells with p-Histone H3 staining was assessed with a minimum of 1559 cells counted from at least 14 different fields for each condition. Adavosertib significantly increased the proportion of BHP7-13 and FTC-133 cells with p-Histone H3 staining. Scale bar, 10 μm.

**Figure 3 cancers-13-03487-f003:**
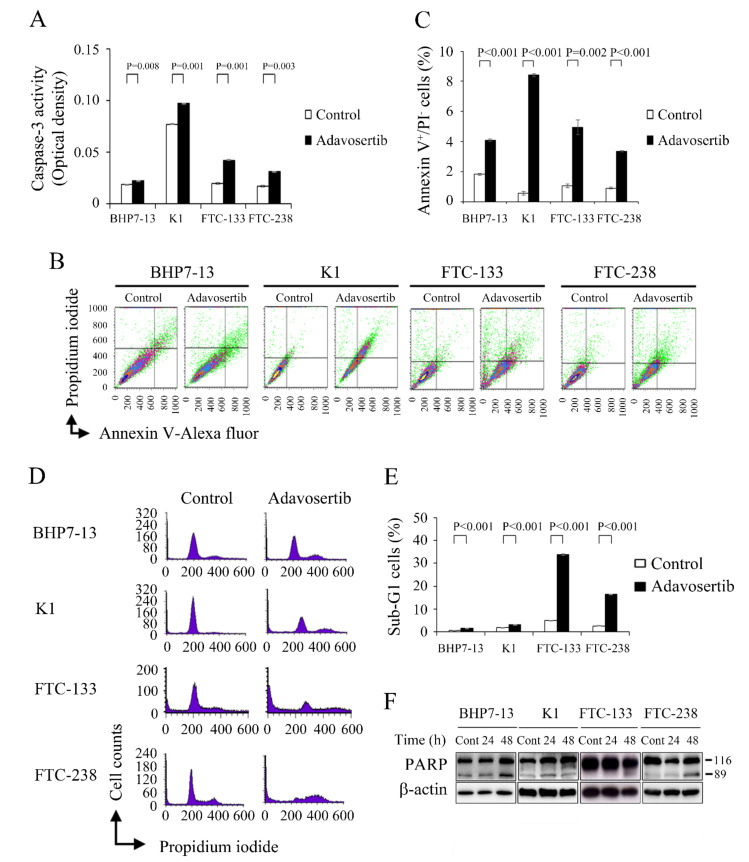
Adavosertib induces apoptosis in BHP7-13, K1, FTC-133, and FTC-238 cells. (**A**) Caspase-3 activity was assessed using a fluorometric assay in DTC cells treated with vehicle or adavosertib (500 nmol/L) for 48 h (BHP7-13 and K1) and 24 h (FTC-133 and FTC-238). Adavosertib treatment significantly induced caspase-3 activity in the four DTC cell lines. (**B**) Early apoptosis was assessed using flow cytometry to detect annexin V-positive/propidium iodide-negative staining in DTC cells treated with adavosertib (500 nmol/L) or vehicle for 48 h (BHP7-13, K1, and FTC-133) and 24 h (FTC-238). (**C**) The statistical analysis of three independent experiments for each condition described in (**B**) showed that adavosertib significantly induced earlier apoptotic cells in the BHP7-13, K1, FTC-133, and FTC-238 cell lines. (**D**) Sub-G1 apoptotic cells were detected by flow cytometry to measure the DNA content in the cells treated with vehicle or adavosertib (500 nmol/L) for 48 h (BHP7-13, K1, FTC-133) and 24 h (FTC-238). (**E**) The statistical analysis from three independent experiments for each condition described in (**D**) revealed that adavosertib significantly increased the proportion of sub-G1 cells in the BHP7-13, K1, FTC-133, and FTC-238 cell lines. (**F**) DTC cells were treated with vehicle or adavosertib (500 nmol/L) for 24 and 48 h. Western blot analysis revealed that adavosertib therapy increased the levels of cleaved PARP, a marker of apoptosis, by 48 h in BHP7-13, FTC-133, and FTC-238 cells and by 24 h in K1 cells.

**Figure 4 cancers-13-03487-f004:**
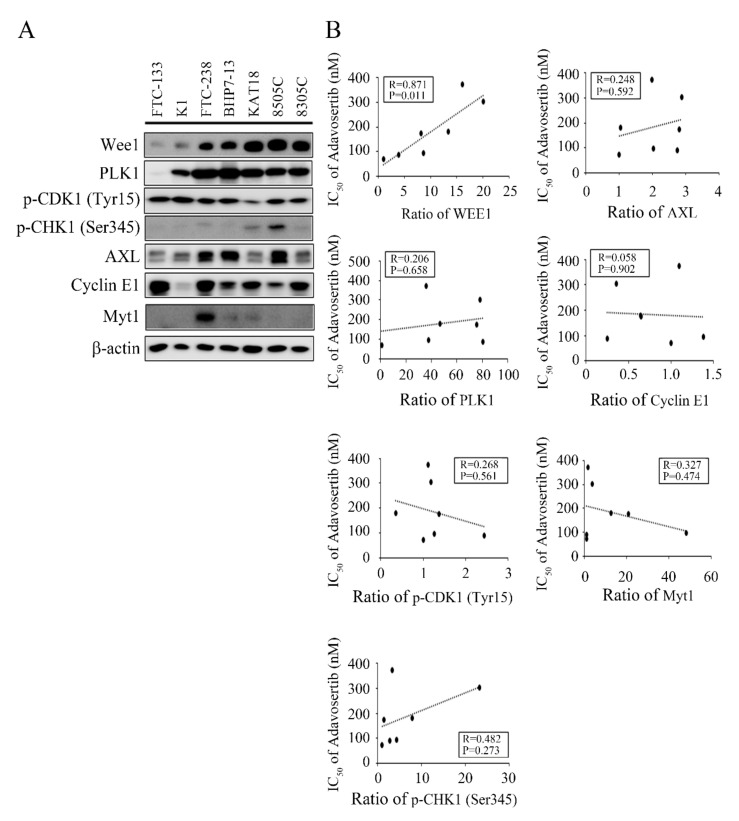
Susceptibility to adavosertib reversely correlates with baseline Wee1 levels. (**A**) The baseline expression of Wee1, PLK1, p-CDK1 (Tyr15), p-CHK1 (Ser345), AXL, cyclin E1, and Myt1 were evaluated by western blot. The loaded proteins were ordered according to the IC_50_ of adavosertib in seven thyroid cancer cell lines. Wee1 expression gradually increased from FTC-133 to 8305C; the expression of the other proteins demonstrated a random distribution. (**B**) Band densities were photographed and quantified. The relationships between IC_50_ and protein level were analyzed using Pearson’s correlation coefficients, and graphs were created using FTC-133 as reference. Wee1 expression had an inverse correlation with sensitivity to adavosertib.

**Figure 5 cancers-13-03487-f005:**
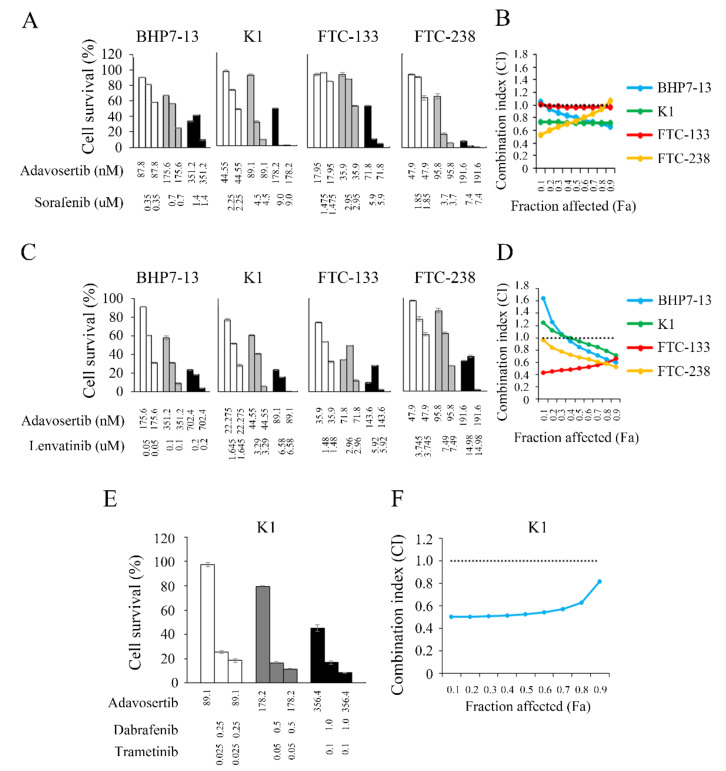
Adavosertib synergistically enhances the cytotoxic effects of sorafenib, lenvatinib, and dabrafenib plus trametinib in DTC cells. (**A**) The cytotoxic effects of adavosertib and sorafenib alone or in combination after a 4-day treatment in BHP7-13, K1, FTC-133, and FTC-238 cells were evaluated using lactate dehydrogenase assays. (**B**) CompuSyn software evaluated the combination index (CI) of adavosertib and sorafenib. The combination of adavosertib and sorafenib had synergistic effects in the K1 cells and synergistic to antagonistic effects in the BHP7-13, FTC-133, and FTC-238 cells. The combination of adavosertib and sorafenib presented synergistic effects in BHP7-13 and FTC-133 when the fraction affected was ≥0.2 and in FTC-238 when the fraction affected was <0.9. (**C**) We employed lactate dehydrogenase assays to determine the cytotoxicity of adavosertib, lenvatinib, and the combination of adavosertib and lenvatinib for a 4-day treatment in the BHP7-13, K1, FTC-133, and FTC-238 cells. (**D**) CompuSyn software assessed the CI of adavosertib and lenvatinib in the four DTC cell lines. Adavosertib plus lenvatinib was synergistic in FTC-133 and FTC-238 and synergistic to antagonistic in BHP7-13 and K1. (**E**) We assessed the cytotoxicity of adavosertib, dabrafenib plus trametinib, and triple combination of adavosertib, dabrafenib, and trametinib for a 4-day treatment in the K1 cells. (**F**) CompuSyn software calculated the CI of adavosertib and dabrafenib plus trametinib. The combination of adavosertib and dabrafenib plus trametinib demonstrated synergistic effects (CI, 0.50–0.82) in the K1 cells. The horizontal dotted line at CI = 1 was drawn to discriminate synergism (CI < 1) from antagonism (CI > 1).

**Figure 6 cancers-13-03487-f006:**
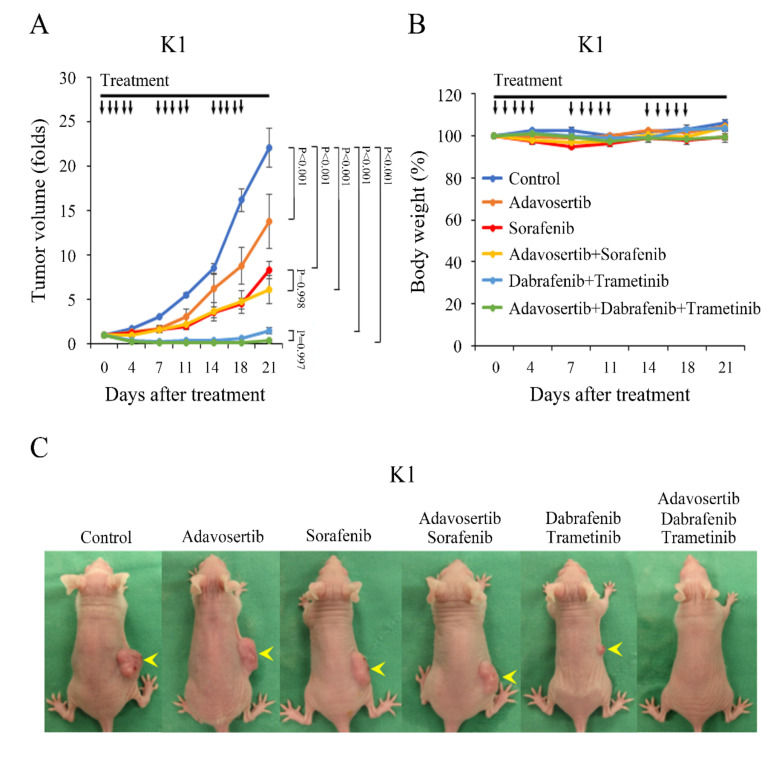
Adavosertib retards subcutaneous xenograft growth in a papillary thyroid cancer model. (**A**) Nude mice bearing K1 xenografts were treated with oral gavage of placebo, adavosertib (50 mg/kg), sorafenib (40 mg/kg), the combination of adavosertib and sorafenib, dabrafenib (30 mg/kg) plus trametinib (0.6 mg/kg), or triple combination of adavosertib, dabrafenib, and trametinib for three cycles of 5 days on and 2 days off. Adavosertib, sorafenib, the combination of adavosertib and sorafenib, dabrafenib plus trametinib, and the triple combination of adavosertib, dabrafenib, and trametinib significantly inhibited K1 tumor growth when compared with vehicle treatment (*p* < 0.001 for both comparisons). Adavosertib and sorafenib combination therapy did not demonstrate superior effects in reducing K1 tumor growth over sorafenib monotherapy (*p* = 0.998). Tumor volumes were also similar between dabrafenib plus trametinib and triple combination therapy (*p* = 0.997). (**B**) Adavosertib, sorafenib, the combination of adavosertib and sorafenib, dabrafenib plus trametinib, and triple combination therapy did not significantly alter body weight by the close of the study. (**C**) Photographs of the representative mice with K1 tumors (arrowhead) were taken on day 15. Arrow, placebo, adavosertib, sorafenib, the combination of adavosertib and sorafenib, dabrafenib plus trametinib, and the triple combination of adavosertib, dabrafenib, and trametinib treatment.

**Figure 7 cancers-13-03487-f007:**
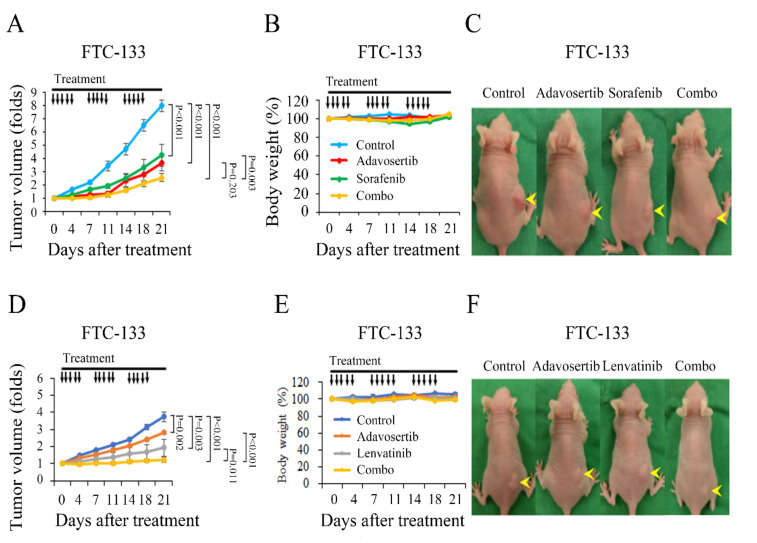
Adavosertib retards subcutaneous xenograft growth in a follicular thyroid cancer model. (**A**) Mice bearing FTC-133 flank tumors were treated with oral gavage of placebo, adavosertib (50 mg/kg), sorafenib (40 mg/kg), or combination therapy of adavosertib and sorafenib daily for three cycles of 5 days on and 2 days off. Compared with control therapy, adavosertib, sorafenib, and combination therapy significantly retarded FTC-133 tumor growth. Combination therapy did not significantly retard FTC-133 xenograft growth when compared with either adavosertib or sorafenib single-agent treatment. (**B**) Adavosertib, sorafenib, and combination therapy did not significantly reduce body weight compared with control treatment. (**C**) FTC-133 xenograft tumors (arrowhead) were photographed on day 21. (**D**) Nude mice with established FTC-133 flank tumors were treated with oral gavage of placebo, adavosertib (50 mg/kg), lenvatinib (30 mg/kg), or combination therapy of adavosertib and lenvatinib daily for three cycles of 5 days on and 2 days off. Adavosertib, lenvatinib, and combination therapy significantly inhibited FTC-133 tumor growth when compared with control treatment. Combination therapy significantly inhibited FTC-133 xenograft growth when compared with adavosertib and lenvatinib single-agent treatment. (**E**) No substantial decreases in body weight were attributable to adavosertib, lenvatinib, or combination therapy compared with control treatment. (**F**) Mice bearing FTC-133 tumors (yellow arrow) were photographed on day 21. Arrow, placebo, adavosertib, sorafenib, lenvatinib, and combination therapy.

## Data Availability

All data were contained within the article and supplementary material.

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
