# Peer review of "Efficacy and Biomarker Analysis of Adavosertib in Differentiated Thyroid Cancer"

_cancers, 2021, doi:10.3390/cancers13143487_

Round 1

Reviewer 1 Report

The authors have presented a paper about "Efficacy and Biomarker Analysis of Adavosertib in Differentiated Thyroid Cancer".

The authors have shown a deep knowledge of the topic and they have presented a very well conducted experiment about a novel drug for DTC.

The results are still not relevent for strictly clinical purposes however I hope that further investigations will allow to understand if this novel drug can be integrated into the clinical practice.

Author Response

We appreciate the reviewer’s very helpful comments.

Our preclinical data reveal the activity of adavosertib in treating DTC with promising safety profiles. Further clinical investigations of adavosertib in treating this disease are indicated. We have addressed this point in revised draft (page 19, lines 580-581).

Reviewer 2 Report

The Authors provide an extensive description of the effects of Adavosertib in DTC in in vitro and vivo models. The results are clearly explained and the design of the work is clear. I have no major questions to ask and I think that the paper deserves publication.

Author Response

We appreciate the reviewer’s thoughtful review of our work and the very helpful comments.

Reviewer 3 Report

Intersting and great work..hope to see future implications.

Author Response

We appreciate the reviewer’s very helpful comments.

Our preclinical data demonstrate the activity of adavosertib in treating DTC with promising safety profiles. Further clinical investigations of adavosertib in treating this disease are indicated. We have addressed this point in revised draft (page 19, lines 580-581).

Reviewer 4 Report

The study is vey well designed and the results are pretty interesting. I have no further comment.

Author Response

(The authors gave the same response as above.)
